# Glucocorticoids Influencing Wnt/β-Catenin Pathway; Multiple Sites, Heterogeneous Effects

**DOI:** 10.3390/molecules25071489

**Published:** 2020-03-25

**Authors:** Katalin Meszaros, Attila Patocs

**Affiliations:** 1Hereditary Tumours Research Group, 1089 Budapest, Hungary; kati.balla@gmail.com; 2Department of Laboratory Medicine, Semmelweis University, 1089 Budapest, Hungary; 3Department of Molecular Genetics, National Institute of Oncology, 1122 Budapest, Hungary

**Keywords:** glucocorticoid excess, Wnt/β-catenin pathway, bone, adipose tissue, brain, skin

## Abstract

Glucocorticoid hormones are vital; their accurate operation is a necessity at all ages and in all life situations. Glucocorticoids regulate diverse physiological processes and they use many signaling pathways to fulfill their effect. As the operation of these hormones affects many organs, the excess of glucocorticoids is actually detrimental to the whole human body. The endogenous glucocorticoid excess is a relatively rare condition, but a significant proportion of adult people uses glucocorticoid medication for the treatment of chronic illnesses, therefore they are exposed to the side effects of long-term glucocorticoid treatment. Our review summarizes the adverse effects of glucocorticoid excess affecting bones, adipose tissue, brain and skin, focusing on those effects which involve the Wnt/β-catenin pathway.

## 1. Introduction

Glucocorticoids (GCs) are pleiotropic hormones with an overall impact on different processes: cell proliferation and survival, growth, reproduction, metabolism and cardiovascular function essential for life. They are also responsible for the proper operation of the immune system and central nervous system [1,2,3]. Glucocorticoid excess at the same time leads to the appearance of severe comorbidities and adverse events [4].

Endogenous glucocorticoid excess, Cushing’s syndrome (CS), is a relatively rare disease, with an annual incidence of 0.7–2.4 cases per million population [5]. Glucocorticoid excess from an exogenous source is much more common, considering that about 1% of the adult population use oral glucocorticoids [6] and the proportion of GC-users in aged persons is even higher [7]. Cushing’s syndrome can be characterized by broad symptomatology, and the clinical appearance of CS often overlaps with features of metabolic, rheumatologic, gynecologic, cardiovascular and neuropsychiatric disorders.

Glucocorticoid signaling engages many signaling pathways to exert its pleiotropic effect in the human body. The most studied pathways include the GH/IGF-1 axis, TGFβ-SMAD signaling, MAPK signaling, PI3/Akt signaling, Wnt-signaling and BMP signaling. All these at various levels can be responsible for signs observed in patients suffering from GC excess.

## 2. Wnt/β-Catenin Pathway or Canonical Wnt-Signaling

The Wnt/β-catenin pathway is an ancient signaling cascade that controls numerous cell processes (stem cell renewal, cell fate determination, differentiation and proliferation) during embryonic development and adult homeostasis [8]. Its deregulation has been detected in a number of human pathologies, including birth defect disorders, skeletal disorders and cancer [9].

Wnt proteins are a family of small signaling glycoproteins, which bind to a member of the Frizzled (Fz) receptor family. In the absence of Wnt-ligand, β-catenin is constantly degraded by a destruction complex composed of Axin and Adenomatous polyposis coli (APC) which admit the phosphorylation of β-catenin by glycogen synthase kinase 3β (GSK-3β) and casein kinase 1 (CK1). Phosphorylated β-catenin is subjected to proteosomal degradation, resulting in low free cytoplasmic and nucleic β-catenin levels in the absence of Wnt-ligand binding. The binding of Wnt to the Fz receptor requires low-density lipoprotein receptor-related-protein 5 or 6 (LRP 5/6) as coreceptors. The formation of the Wnt-Fz-LRP 5/6 complex leads to the membrane recruitment of the Axin complex by Dishevelled proteins (Dvl), Axin being bound to a cytoplasmic tail of LRP 5/6. In the absence of the degrading complex, β-catenin escapes from phosphorylation and consequential ubiquitination. This leads to an increase in free β-catenin levels which form complexes in the nucleus with the DNA-bounded T cell factor/lymphoid enhancer factor (TCF/LEF) and activate Wnt target gene expression. 

Up to now,10 families of Wnt-pathway agonists were identified [10], from which the role of the Dickkopf family (Dkks), sclerostin (Sost) and secreted Frizzled-related proteins (sFRPs) were studied in depth in relation with glucocorticoid excess (Figure 1).

## 3. Wnt/β-catenin Pathway and Cushing’s Syndrome

The common occurrence of adrenal adenomas in patients with familial adenomatous polyposis highlighted the role of the Wnt/β-catenin pathway in adrenal tumorigenesis [14,15]. With APC being part of the degradation complex of β-catenin, abnormal Wnt/β-catenin activation was observed in both benign and malignant adrenocortical tumors: in 10 of 26 adrenocortical adenomas (38%) and in 11 of 13 adrenocortical carcinomas (77%) [16]. In these specimens, immunohistochemistry revealed atypical nuclear and/or cytoplasmic β-catenin accumulation. In addition, 27% of benign and 31% of malign tumors represented activating somatic mutations of the β-catenin gene (CTNNB1), altering the Ser45 of exon 3 [16], involved in targeted degradation of β-catenin [17].

In another study, genetic alterations were found in 5 of 33 adrenal adenomas (15%). Of these, three were cortisol-secreting, one aldosterone-producing and one nonfunctioning adenoma [18]. In two samples, point mutations were demonstrated at serine residues of codons 37 and 45, another three samples contained deletions [18]. There are also results suggesting that adrenocortical tumors with molecular alterations of CTNNB1 are primarily nonsecreting adenomas (61.1%), but the role of the Wnt/β-catenin pathway in adrenocortical tumor size and cortisol production seems plausible [19].

## 4. Effects of GC Excess on Bone

### 4.1. Bone Homeostasis and Main Cytokines

Homeostasis of the adult skeleton depends on the balance between bone-forming and bone-resorbing processes and relies on the complex communication between osteoblasts, osteocytes and osteoclasts.

Bone-forming osteoblasts are derived from mesenchymal stem cells (MSCs) located in the bone marrow. Master transcriptional regulator Runx controls the commitment of MSCs toward osteogenesis and the following differentiation processes [20]. Preosteoblasts are subjected to a maturation process resulting in matrix-synthesizing osteoblasts. Osteoblasts trapped in the mineralized matrix become osteocytes, the most substantial and long-lived cell types of the bone which communicate with each other through their cytoplasmic extensions. Osteocytes are involved in the bone turnover process through their mechanosensory activity, controlling the operation of both osteoblasts and osteoclasts [21]. Osteoclasts originate from hematopoietic stem cells and beside their “classic” function of bone resorption, the findings of recent years revealed their complex role in bone homeostasis, hematopoiesis and angiogenesis [22].

The communication between bone cells is conducted partly through the production of cytokines. Osteoblasts release M-CSF, which binding to its cognate receptor on preosteoclasts, provides signals essential for the survival and proliferation of osteoclast precursors [23]. The receptor activator of nuclear factor kappa-B°(RANK) is expressed in osteoclast precursors and osteoclasts; intracellular RANK signaling results in the induction of transcription factors required for osteoclast differentiation. RANK ligand (RANKL) is expressed mainly in osteoblasts, osteocytes and bone stroma [24]. RANKL–RANK interaction is critical for the differentiation of osteoclast precursors to mature osteoclasts [25]. Osteoprotegerin (OPG) is produced by the cells of osteoblast lineage, mainly by osteocytes [26]. Osteoprotegerin is a decoy receptor for RANKL, thereby inhibiting the differentiation and activation of osteoclasts.

### 4.2. Detrimental Effects of GC Excess

Glucocorticoid-induced osteoporosis (GIO), the most common cause of secondary osteoporosis [27,28], is characterized by the decrease of bone mass and microarchitectural deterioration which results in increased bone fragility [28]. As it was summarized in detail by Komori, GC excess exerts its detrimental effect on bone formation by inducing apoptosis of osteoblast and osteocytes, increasing production of reactive oxygen species and inhibiting the Wnt-signaling pathway [29]. Related to bone resorption, excess of GCs influences RANKL and OPG production besides prolonging the life span of osteoclasts [29]. The Wnt/β-catenin pathway is involved in all of these processes.

### 4.3. Inadequate Wnt/β-Catenin PathwayOperationdue to GC Excess

GC excess rapidly induces the apoptosis of osteoblastic cells as it was demonstrated on different cell lines and in patients with CS or receiving GC treatment. Sclerostin treatment of human osteoblastic cells initiates apoptosis through a caspase-activated mechanism [30]. sFRP-1 overexpression in human osteoblasts accelerates apoptotic cell death in osteoblasts and osteocytes [31]. These findings were also confirmed in humans: in adult patients on steroid therapy, biochemical markers were assessed at baseline and at 24, 48 and 96 h during the treatment. Osteocyte-related biochemical markers OPG and sclerostin showed a significant decrease at 48 h (after which their level was practically unchanged during the rest of the experiment); it can be explained by a rapidly induced apoptotic cell death of osteocytes [32]. Low serum sclerostin level was also found in patients with Cushing’s syndrome, but during treatment of GC excess, this level increased [33].

Glucocorticoids are detrimental to osteo-anabolic processes through their effects on oxidative stress levels [34,35]. The administration of prednisolone for 28 days to mice resulted in increased reactive oxygen species (ROS) levels in the bone marrow [34]. In human osteoblasts, dexamethasone significantly increased the number of mRNA transcripts related to oxidative stress [35]. The increase of ROS leads to enhanced activity of Forkhead box O (FoxO) transcription factors [34]. The interaction between FoxOs and β-catenin prevents β-catenin from forming complexes with TCF/LEF and activates Wnt target gene expression [36]. Diverting the limited pool of β-catenin from β-catenin-TCF/LEF mediated transcription to β-catenin-FoxO transcription itself is a significant proof for the repression of bone-anabolic processes in GC excess. In line with this, Iyer et al. demonstrated that FoxOs attenuated Wnt-signaling in lineage-committed murine osteoblast progenitor cells, hence decreasing the number of matrix-synthesizing osteoblasts [37].

Glucocorticoid excess enhances the expression of Wnt-inhibitors. Supraphysiological levels of glucocorticoids enhance sFRP-1 expression in primary mesenchymal cell cultures in vitro and in osteoblasts and bone marrow cells in vivo [38]. The expression of another Wnt-inhibitor, Dkk-1, increased after dexamethasone treatment in primary human osteoblasts [39]. Sato et al. demonstrated that glucocorticoids increase the expression of Sost mRNA and the number of sclerostin-positive osteocytes in wild-type mice while, in sclerostin deficient mice, the negative effects of GCs on bone (decreased mass, deteriorated microarchitecture, reduced structural and material strength) were prevented [40].

Wnt-pathway inhibition influences cell proliferation and maturation of cells from osteoblastic lineage. Dkk-1 expression in rat calvaria cells was associated with inhibited osteoblast differentiation and with the lack of mineralized nodules; parallelly, increased adipocyte differentiation could be detected in vitro [41]. In Yao’s experiment, expression levels of Lrp5 and Lrp6 (coreceptors of Wnt/β-catenin signaling), Runx, Osterix and osteocalcin (osteoblast differentiation and maturation markers) and unphosphorylated β-catenin were assessed in sFRP-1 transgenic and wild-type male and female mice. Lrp5 and Lrp6, β-catenin and osteocalcin levels were significantly lower in both sexes of sFRP-1 Tg mice, while Osterix and Runx (osteoblast differentiation markers) were significantly lower only in sFRP-1 male mice compared to controls [13]. These results suggest an important role for Wnt/β-catenin pathway in osteoblast differentiation and maturation, and also highlight the gender differences in GC-induced bone lesions.

The loss of osteocytes by GC-induced apoptosis damages the osteocyte-canalicular network and the mechanosensory ability of bone, which, in turn, decreases the capacity of osteocytes to regulate osteoclasts’ function. In mechanosensory osteocytes, the capacity to produce RANKL is much higher than that in osteoblasts [42]. Mice lacking RANKL in their osteocytes were protected from the adverse effects of glucocorticoid treatment on bone resorption [43].

Intact Wnt-signaling exerts an indirect negative role on osteoclastogenesis: in pluripotent mesenchymal cells, and stromal cells Wnt3a upregulates OPG expression [44], thereby inhibiting RANKL–RANK interaction and consequently the differentiation of octeoclasts. Wnt-pathway inhibitor sclerostin upregulates the expression of RANKL and downregulates OPG expression in human primary preosteocytes and murine osteocyte-like MLO-Y4 cells [45]. Dkk-1 enhances osteoclastogenesis by increasing the expression of RANKL and M-CSF and downregulating OPG in mesenchymal progenitor cells undergoing osteoblastic differentiation [25]. In another study, GC treatment of cortical bone organ cultures and primary osteoblasts did not increase significantly RANKL, but suppressed OPG production, suggesting that cortical bone loss caused by glucocorticoid excess can be attributed more likely to reduced OPG rather than elevated RANKL expression [43].

The role of sFRP-1 in the regulation of osteoclasts now seems to be bidirectional: first, sFRP-1 inhibits osteoclast formation by preventing the fusion of osteoblast precursors [46], secondly, sFRP-1 increases osteoclast surface in sFRP-1 transgenic female mice by 38% compared to WT controls [13].

Recently, Albers et al. demonstrated a direct effect of the Wnt-signaling pathway on osteoclastogenesis and bone resorption beside the indirect effects mediated by osteocytes [47]. They found elevated Frizzled8 (Fzd8) expression in osteoblasts and osteoclasts. The skeletal phenotype of Fzd8-deficient mice showed osteopenia with unaffected bone formation and enhanced osteoclastogenesis. To assess whether this is an indirect effect mediated by osteoblastic cytokines or it is the direct effect of the canonical Wnt-pathway on osteoclasts, they examined mice lacking ß-catenin in their osteoclast lineage. It was found that these animals displayed increased bone resorption despite their normal OPG level produced by osteoblasts [47].

In summary, we can conclude that the Wnt/β-catenin pathway is essential in normal bone homeostasis, and the harmful effects of GC excess are closely related to the inadequate operation of this pathway.

## 5. Effects of GC Excess on Adipose Tissue

### 5.1. Adipogenesis and Wnt-Signaling

Adipogenesis is a well-orchestrated process during which MSC-derived preadipocytes develop into mature adipocytes. This process involves the interplay of several transcription factors, from which PPARγ functions as a master regulator [48].

There is plenty of evidence for strong communication between adipose tissue and bones, based (partly) on the interaction between PPAR- and Wnt-signaling pathways. Sustained activation of the Wnt/β-catenin pathway by expression of Wnt10b and inhibiting expression of PPARγ-associated adipogenic genes, in murine preadipocytes, maintains preadipocytes in an undifferentiated state [49]. Inversely, PPARγ induces the decrease of β-catenin expression during the differentiation process of preadipocytes into adipocytes [50]. A β-catenin binding domain within PPARγ ensures the close cooperation between the two pathways [51] and activation of PPARγ induces the proteasomal degradation of β-catenin which is preceded by its phosphorylation by GSK-3β [49]. LiCl, an inhibitor of GSK-3β, prevents PPARγ-induced decrease of β-catenin [49]. Missense mutations of the *Wnt10b* gene detected in families with early-onset obesity represent additional evidence for the strong relation between PPAR- and Wnt-signaling [52].

### 5.2. Detrimental Effects of GC Excess

Clinically, the strong but opposite-sign communication between adipogenesis and osteoblastogenesis appears most expressively in patients with Cushing’s syndrome presenting with osteoporosis, osteopenia and increased bone fragility associated with central obesity [27,53]. GC excess leads to abdominal, especially visceral, fat accumulation accompanied by the depletion of peripheral subcutaneous fat depots. GC-induced fat distribution can be partly explained by the different adipogenic and lipolytic effects of glucocorticoids on visceral adipose tissue (VAT) and subcutaneous adipose tissue (SAT). Analyzing the up- and downregulated genes during dexamethasone treatment of omental and abdominal subcutaneous tissues of obese women revealed that the PPAR signaling pathway was significantly upregulated in both types of adipose tissue but expression levels were higher in VAT than in SAT [54].

GCs change the characteristics of adipose tissue differently in VAT and SAT, as it was demonstrated in methylprednisolone-treated rats [55]. An increased number of adipocytes was experienced both in SAT and VAT, but the size of adipocytes was unchanged in VAT and decreased in SAT. Adipose mass of GC-treated rats increased only in VAT and was practically unchanged in SAT compared to controls. Importantly, GCs inhibit Wnt/β-catenin pathway both in SAT and VAT: Wnt10b and β-catenin levels were significantly decreased, while the Dkk-1 level was significantly increased in methylprednisolone-treated rats. GCs were also proved to exert a different lipolytic effect on VAT and SAT: in SAT, methylprednisolone induced a higher expression of lipolytic enzymes compared to untreated controls, in VAT a GC-induced, suppressive effect on lipolysis was demonstrated [55]. Analyzing abdominal and femoral adipose tissues in patients with Cushing’s syndrome leads to similar results: abdominal adipose tissue was characterized by enlarged fat cells, increased adipose tissue lipoprotein lipase activity and low lipolytic capacity [56].

### 5.3. The Consequences of Adiposity on Wnt-Signaling 

Adipose tissue produces Wnt-inhibitors belonging to the sFRP family. Gene expression of all sFRPs (sFRP1-5) was analyzed in subcutaneous and visceral adipose tissues of obese and non-obese women [57]. In SAT, sFRP-1 was decreased and sFRP-2 increased in obesity and accordingly, sFRP-1 negatively, while sFRP-2 positively associated with insulin resistance [57]. Partly contradictory, another study demonstrated a biphasic regulation of sFRP-1 in mice (confirmed also in human tissues) fed with a high-fat diet. A higher expression of sFRP-1 in mild obesity and with a gradual fall of sFRP-1 in morbidly obese individuals was reported [58]. sFRP-1 overexpression promoted adipogenesis in preadipocytes through the inhibition of the Wnt/β-catenin pathway, suggesting its role in the paracrine regulation of adipogenesis [58].

Adipocytes secrete a variety of signaling proteins (called adipokines) and proinflammatory cytokines which are responsible for the auto-, para- and endocrine effects of adipose tissue. Adipose tissue-derived cytokines (leptin, chemerin, adiponectin, omentin, resistin, IL-6 and TNF-α) were proved to affect the function of osteoblasts and/or osteoclasts [26,59] and not surprisingly, many of these signaling molecules engage with the Wnt/β-catenin pathway.

Leptin is one of the first discovered adipokines and nowadays it is considered as the prototype of all signaling molecules produced by the adipose tissue. It is responsive to extensive communication between adipose tissue and brain, pancreas and reproductive organs [60]. The relation between obesity, leptin levels and Wnt-signaling was studied in depth. Chen et al. showed that activation of the Wnt-pathway, besides repressing preadipocytes’ differentiation, stimulates leptin production in mature murine adipocytes [61]. Benzler et al. revealed that GSK-3β (responsible for the continuous degradation of β catenin) activity was increased in mediobasal hypothalamic cells in obese male mice; the consequence of this status was hyperphagia with further aggravation of glucose intolerance and obesity [62]. Furthermore, they found that genes involved in the Wnt-pathway were mostly expressed in mediobasal hypothalamic cells with particular prominence in the arcuate nucleus. In brain sections of obese, leptin-deficient mice Wnt antagonist Dkk-1 level was significantly higher than in wild-type mice [62]. Leptin deficiency was proved to lead to decreased activation of LRP 6 (co-receptor of Wnt activation); the lower level of activated LRP 6 was fully restored after leptin administration. Leptin treatment, at the same time, caused inhibition of GSK-3β in orexigenic neuropeptide Y (NPY) cells [63]. Very recently, Wang et al. demonstrated the mediator role of neuropeptide Y in glucocorticoid-induced osteoporosis and bone marrow adiposity. In mice, NPY deletion was protective against glucocorticoid-induced loss of bone mass and fatty marrow development [64]. These findings altogether suggest that adipose tissue through leptin expression communicates bilaterally with mediobasal hypothalamic cells and provides evidence for the role of the Wnt/β-catenin pathway in the neuroendocrine control of metabolism (Figure 2).

## 6. Effects of GC Excess on Brain

### 6.1. Neurogenesis and Wnt/β-Catenin Pathway

Data from the last two decades have provided sufficient evidence for the role of Wnt-signaling in different processes of neurogenesis. Recently, Guo summarized these results and concluded that the Wnt/β-catenin signaling pathway plays a context-dependent role in oligodendrogenesis, oligodendrocyte differentiation and myelination. In addition, the effects associated with the Wnt/β-catenin pathway depend on the developmental stage of CNS, oligodendrocyte and CNS microenvironment [65].

### 6.2. Detrimental Effects of GC Excess

Detrimental effects of increased GC levels on brain structure, neurological function and emotional life have been demonstrated in numerous studies. GC excess negatively affects memory retrieval, cognitive functions and behavior and they have significant effects on brain structure [66,67].

Of brain regions, the hippocampus seems to be the most vulnerable to the neurotoxic effect of GCs [67,68,69]. Mice exposed to chronic stress showed reduced hippocampal volume with reduced neurogenesis in the dentate gyrus and neuronal loss and dendritic atrophy in the cornu ammonis 1 (CA1) region [68]. Besides, these animals showed increased hippocampal levels of Dkk-1 and reduced expression of β-catenin.

Clinically, brain development and affective problems of preadolescent children with or without fetal exposure to synthetic GCs were evaluated by Davis et al. [70]. A significant bilateral cortical thinning, particularly in the rostral anterior cingulate cortex was detected in children exposed to prenatal GC treatment. Besides this, the left rostral anterior cingulate cortex was thinner in children with affective problems, underlining the effect of GCs on behavioral disorders occurring even at later ages [70]. MRI results of the data of 339 CS patients lead to the conclusion that these patients have smaller hippocampal volumes, enlarged ventricles and cerebral atrophy; importantly, a part of structural deviations persisted even after a long-term remission [67].

Accumulating evidence suggests that long-lasting effects of GCs on the brain are regulated through epigenetic mechanisms. Rat embryonic neural stem cells exposed to dexamethasone underwent heritable alterations; GC effects induced in parent cells were detectable in daughter cells which were never exposed directly to GCs [71]. Methylation studies showed a significant decrease in global DNA methylation both in parent and daughter cells. Furthermore, dexamethasone caused a long-lasting, heritable receptivity of neural stem cells to oxidative stress manifested in increased apoptotic cell death of dexamethasone-treated daughter cells compared to control cells [71].

Extensive genome-wide alterations in promoter methylation, typically the hypomethylation of the affected promoters, were detected in the hippocampi of fetal guinea pigs treated prenatally with GCs [72]. The samples were collected 24 h and 14 days after two courses of maternal GC treatment. Twenty-four hours after the treatment, hypomethylation and hyperacetylation of the affected genes could be detected. After 14 days of GC administration, the changes previously observed did not persist, but the hypo- or hypermethylation of other promoters was noticed instead. These findings suggest a dynamic, spillover effect of GCs on the prenatal brain [72].

### 6.3. GC Excess and Wnt-Signaling

Inhibitory effects of dexamethasone on the neural precursor cells and the role of the Wnt/β-catenin pathway in this process were described by Boku et al. [73]. The negative effect of GCs has turned aside through the negative regulation of GSK-3β (by the administration of lithium or valproic acid, both inhibiting GSK-3β activity) and the consequential upregulation of Wnt/β-catenin signaling pathway [73,74]. Contrary to that, transgenic mice expressing a constant β-catenin level in neural precursor cells had enlarged brains with increased cerebral cortical surface and enlarged lateral ventricles [75].

The negative effects of GCs were demonstrated on human neural progenitor cells (hNPCs) responsible for the major cell types (neurons, astrocytes and oligodendrocytes) of the brain. Treating hNPCs with dexamethasone caused a marked decrease in proliferation and neuronal differentiation of hNPCs while increased glial cell formation. Dexamethasone induced the upregulation of the Wnt-signaling inhibitor Dkk-1 which effect could be inhibited by adding a GR antagonist, mifepristone. The final conclusions were that Dkk-1 was the primary target for GR in hNPCs and the regulation of hNPCs by dexamethasone and Dkk-1 was mediated through the canonical Wnt-signaling pathway [76].

High levels of GCs induced the generation of reactive oxygen species (ROS) in primary hippocampal and cortical cell cultures [77]. Accordingly, GC excess increased the level of oxidized components and decreased the antioxidant enzymes including SOD, catalase and glutathione peroxidase in the hippocampus of young rats [78]. Oxidative stress in rats caused hippocampal oxidative damage and apoptosis of pyramidal cells with a significant decrease in GRs in the hippocampal CA1 region and a consequential decline in cognitive functions [69]. In neonate rats, postnatal glucocorticoid treatment caused a significant decrease in total brain volume and neuronal loss in the dentate gyrus and CA1 region, which effects were largely prevented when dexamethasone therapy was combined with antioxidant vitamins [79]. Besides, sharing β-catenin as a common co-activator, the crosstalk between FoxO-regulated pathways and Wnt-signaling in neurogenesis was also confirmed by the identification of Wnt-inhibitors as direct targets of FoxOs. In FoxO-null neural stem cells, a significant downregulation of sFRP-1, sFRP-2 and sclerostin was demonstrated [80].

## 7. Effects of GC Excess on Skin

### 7.1. Skin and Steroidogenesis

Skin, besides its primary function of protecting the body, is an important part of immunological processes and extraadrenal steroidogenesis [81]. Recently, it was found that skin has its own HPA axis analog system with all major components present in the HPA axis and enzymes required for glucocorticoid synthesis [81,82].

Keratinocytes, one of the main cell types of the epidermis, synthesize cholesterol, the precursor of all steroids [83]. Epidermal and follicular keratinocytes, melanocytes, hair follicles and dermal fibroblasts have been proven to synthesize cortisol [82,84]. Glucocorticoids act through glucocorticoid receptors which are present in most, if not all of the skin compartments [84]. Skin expresses both types of 11-beta-hydroxysteroid-dehydrogenases (11β-HSDs); type 1 enzyme activates cortisone into cortisol, while type 2 converts cortisol into inactive cortisone [85]. The balanced activity of these enzymes is essential in the regulation of locally produced or exogenously administered glucocorticoids.

### 7.2. Detrimental Effects of GC Excess

Related to Cushing’s syndrome, a part of the signs and symptoms are relatively common (fatigue, weight gain, insomnia, irritability, depression, etc.) and makes CS difficult to diagnose. Some of the skin-related manifestations (easy bruising, reddish-purple striae) are between those signs which best discriminate Cushing’s syndrome from other diseases, while other skin-related signs (thin skin, poor skin healing) are very common in patients with CS (70%, respectively 80%) [86].

GCs have profound effects on the different layers of the skin. In the epidermis, the decreased size of keratinocytes, the thinning of the epidermis and the diminution of the lamellar lipid bilayer in the stratum corneum are well-known histopathological features of GC excess [87]. The molecular mechanism behind the “thinning of the epidermis” is that GCs inhibit the proliferation and migration of keratinocytes and have a dual effect on keratinocytes’ differentiation: at early stages of differentiation, inhibitory, while in late stages, stimulatory [88]. Transcriptional profiles of human keratinocytes untreated and treated with dexamethasone revealed that (unlike in many tissues affected by GCexcess) GCs exert anti-apoptotic effects on keratinocytes by inducing anti-apoptotic genes and suppressing proapoptotic genes [88].

In the dermis, GCs lead to a decreased number of fibroblasts with the rearrangement of the dermal fibrosis network [87]. The antiproliferative effect of GCs on fibroblast results in an altered collagen and elastin synthesis and in the thinning of the dermis, which causes a decrease of mechanical strength and elasticity of the skin.

There are only little data about the effect of GC excess on hypodermis, but we have to note the confusion in the literature related to the definition of skin-associated adipocyte layers. Traditionally, the thick layer of adipocytes underlining the reticular dermis has been called hypodermis or subcutis and it was considered that all skin-associated adipose tissues represent one depot [89]. The presence of a common precursor for dermal fibroblasts and intradermal adipocytes, besides the independent development of skin-related adipocytes and subcutaneous adipose tissue [90,91] justifies Driskell et al.’s proposal for a refined nomenclature. They recommend the term of intradermal adipocytes and dermal white adipose tissue for cells and adipose tissue underlying the reticular dermis [89].

Related to the effects of GC excess on skin compartments, it seems plausible that skin atrophy involves all skin layers, including dermal white adipose tissue. An indirect proof for this assumption was presented very recently by Baida et al. [92]. They studied the expression of FK506-binding protein 51 (FKBP51), a co-chaperone and regulator of the glucocorticoid receptor, in mice and human skin. They found that all skin compartments: epidermis, dermis, dermal adipose and CD34+ stem cells of FKBP51 KO mice were much more resistant to GC-induced hypoplasia than of their wild-type littermates [92].

Epidermal appendages are also influenced by GC excess. Based on the data of 481 patients published by the European Registry on Cushing’s syndrome, hair loss was present in 31% of patients [93]. Female balding occurs five times more often in patients with CD than in age-and sex-matched controls; the combination of female balding with the symptom of weakness/fatigue was 15 times more common in CD patientsthan in controls [94]. Rarely, female baldness is the main complaint of patients with Cushing’s syndrome [95].

### 7.3. Wnt-Pathway Alterations and Their Consequences

Ectopic expression of Wnt-inhibitor Dkk1 in epidermis drives to the lack of epidermal appendages: hair, whiskers, teeth and mammary gland [96]. Epithelial β-catenin deletion or ectopic Dkk-1 expression, induced during the growth phase of the hair cycle, causes premature hair follicle regression [97]. Another Wnt-inhibitor, sFRP-1, also plays a role in hair loss: WAY-316606, an sFRP-1 inhibitor, enhanced hair shaft production, hair shaft keratin expression and inhibited spontaneous hair follicle regression in organ-cultured human scalp ex vivo [98]. In line, a conditional gain-of-function mutation of β-catenin resulted in accelerated and excessive hair generation with densely packed hair follicles and little or no interfollicular epidermis [99].

As the canonical Wnt-signaling plays an important role in skin cells’ proliferation and differentiation, its impact on wound healing is self-evident. Wnt10a-deficient mice, besides their phenotype including alopecia and follicle growth failure, showed delayed wound healing compared to their wild-type littermates [100]. De novo development of hair follicles after injuries is also driven by Wnt-signaling: inducing expression of Dkk-1 after wounding leads to the inhibition of hair follicle neogenesis in mice [101]. Stabilization of nuclear β-catenin caused delayed wound healing in human skin organ cultures: LiCl-treated wounds (LiCl inhibits GSK-3β, an inactivator of β-catenin) showed a significantly larger wound size on day four after wounding compared to untreated wounds (12% healing rate for LiCl-treated wounds vs. 70% healing rate for untreated wounds) as LiCl treatment inhibited the migration of primary human keratinocytes in wounds [102].

In non-healing edges of venous ulcers, components of the Wnt-pathway were found to be deregulated: phosphorylated β-catenin was found mainly in the cytoplasm and nuclear; while in healthy skin specimens, β-catenin was present mostly at the membrane of the basal layer [103]. Jozic et al. demonstrated that GCs promote nuclear localization of β-catenin: dexamethasone treatment leads to a robust nuclearization of β-catenin in primary human keratinocytes, of which the effect could be blocked by pretreatment with mifepristone (RU486), a glucocorticoid receptor antagonist [104].

We have to conclude that, related to GC excess-induced skin alterations, the sole role of the Wnt/β-catenin pathway has not yet proven in many processes, but its function obviously contributes to these phenotypes. Future research can clarify the bidirectional link between the Wnt-pathway and GC-induced skin lesions.

## 8. Future Perspectives

Balanced operation of the Wnt/β-catenin pathway is indispensable for homeostatic processes in many tissues, including bones, brain, adipose tissue and skin (Figure 3.). However, some missing details related to its extensive impact on cell differentiation and proliferation and to its crosstalk with other pathways suggest that regulating its operation may be hazardous and it can be only done with the utmost care. In bone, treating osteoporotic and osteolytic bone lesions and treating myeloma multiplex with compound regulating Wnt-inhibitors showed some encouraging results. In myeloma cells, Dkk-1 is produced in large amounts and it is overexpressed in patients having extensive osteolytic lesions [105]. Bortezomib, a proteasome inhibitor approved for the treatment of relapsed multiple myeloma patients was proved to inhibit Dkk-1 expression in neonatal mouse calvarial cells and bone marrow stromal cells [106]. In another study, Wnt-pathway independent β-catenin activation and consequential osteoblast differentiation were demonstrated, suggesting a role for bortezomib in the partial restoration of tumor-induced suppression of Wnt-signaling [107].

Selective inhibition of sFRP-1 was also proposed as a potential tool to stimulate Wnt-signaling, thereby osteo-anabolic processes in bone [13,108]. In certain types of cancer, epigenetic inactivation of different sFRP genes was identified, thus sFRPs were hypothesized to function as tumor suppressors [109,110]. These data underline the necessity of strict controls and a careful evaluation of results obtained during experiments related to Wnt-inhibitors.

Wnt/β-catenin signaling can be stimulated by physical exercise. In a study conducted among women affected by abdominal obesity, osteoblast cell cultures were exposed to sera of these women at an initial timepoint and 4, 6 and 12 months after they attended a physical training program combined with a hypocaloric diet [111]. Proteins of the Wnt/β-catenin pathway were quantified at the specified timepoints. Results showed a time-dependent restoration of the Wnt/β-catenin pathway evident already after the fourth month of training. At the same time, a consequential decline in the inhibition of the pathway was demonstrated by decreased sclerostin levels. These results suggest that the decrease of the abdominal fat mass due to physical training and hypocaloric diet created a hormonal environment where osteoblasts restored their osteo-anabolic function [111].

A similar study was carried out among breast cancer survivors, where serum levels of Dkk-1 and sFRP-1 were followed before and after a 12-week long physical exercise training program. The conception of this study was based on the results of former studies showing that elevated serum Dkk-1 levels correlated with poor prognosis in certain types of cancer [112]. Significant decrease in serum levels of Wnt-inhibitors: Dkk-1 and sFRP-1 were demonstrated after the exercise training program, accompanied by the significant decrease of serum insulin and leptin levels and significant reduction of total body fat and visceral fat area [113].

Another evidence for the beneficial effects of physical exercises on Wnt-pathway was the mechanosensitive expression of sclerostin (specific product of osteocytes). Mechanical unloading of wild-type mice caused downregulation of Wnt/ß-catenin signaling with the upregulation of the sclerostin levels [114]. In Sost KO mice, unable to produce sclerostin, bone loss caused by mechanical unload could not be detected [114]. This finding was also confirmed in humans: serum sclerostin level was assayed in healthy adult men before and after longer bed rest and increased sclerostin level wasdetected. Lower serum sclerostin level was accompanied by significantly increased bone resorption markers (bone formation markers were unchanged), with significantly higher urinary calcium excretion and significantly declined bone mineral density at weight-bearing skeletal sites: lumbar spine, hip, femoral neck and calcaneus [115]. These results suggest that the simultaneous presence of glucocorticoid excess and mechanical unload (e.g., sedentary lifestyle) exert a cumulative negative effect on bone homeostasis.

The beneficial effects of physical exercise were also highlighted by Chen, who recently reviewed the paradox of chronic stress and chronic exercise, both known to increase basal cortisol levels. Their metanalysis underlined that solely increased basal cortisol levels are not necessarily detrimental to the human body and their work suggested that other factors are also needed for the harmful effects of glucocorticoid excess [116]. Related to GIO, Tóth et al. highlighted that one of the most peculiar features was the large interindividual variability in clinical presentation and severity [27]. In this regard, the role of hypoxia and hypoxia-induced factors seem unavoidable. In bone, despite the detailed investigations, the effect of hypoxia, whether it is osteo-anabolic or osteo-catabolic is not entirely clear. A proof for the role of hypoxia in GC excess induced detrimental effects was shown recently by Ueda. In MLO-Y4 murine osteocytic cells, dexamethasone and hypoxia separately caused apoptosis; but their simultaneous effects lead to a significant increase of necrotic celldeath [117].

Related to this, Noguchi et al. proved that the detrimental effects of the glucocorticoid excess on bone could be prevented by oxygen ultra-fine bubbles saline injection [118]. In prednisolone-treated mice, regularly administered injections diluted with oxygen ultra-fine bubbles prevented severe bone loss caused by GCs. Simple saline injections and injections diluted with nitrogen ultra-fine bubbles were not able to counteract the detrimental effects of GCs. In vitro experiments revealed that appropriate oxygen supply (through injection with oxygen ultra-fine bubbles) had not affected differentiation of osteoblasts, but inhibited osteoclastogenesis, resulting in an osteo-anabolic effect [118].

Summarizing these results, it can be assumed that the restorative effects of physical exercise on Wnt/ß-catenin pathway can be linked partly to an enhanced oxygen supply, and inversely, deleterious effects of GC excess on Wnt-pathway are more pronounced when hypoxic conditions are present and probably the severity of the damaging effects is related to the degree of hypoxia.

The relationship between GC excess and severe comorbidities is well-known: fractures occur in 30%–50% of the patients subjected to long-term hypercortisolism [28]; the rate of glucocorticoid-induced hyperglycemia or diabetes among chronic GC-users is more than 32% [119]. Besides this, glucocorticoids are responsible for cardiovascular (15%), infectious (15%), gastrointestinal (10%), psychological/behavioral (9%) and endocrine/metabolic (7%) adverse effects [4]. Therefore, the earlier treatment of endogenous GC excess is imperious and the periodic evaluation of chronic GC-users’ state is indispensable. This is also of paramount importance because a part of the adverse effects of GC excess seems to be long-lasting, definitive or even heritable. Besides the complex medical treatment of endogenous GC excess and the adverse effects of GCtreatment, the role of comprehensive lifestyle changes are also essential: physical activity and the management of obesity create a hormonal environment in which Wnt/β-catenin pathway can fulfill its role in the homeostatic processes of the adult organism.

## Figures and Tables

**Figure 1 molecules-25-01489-f001:**
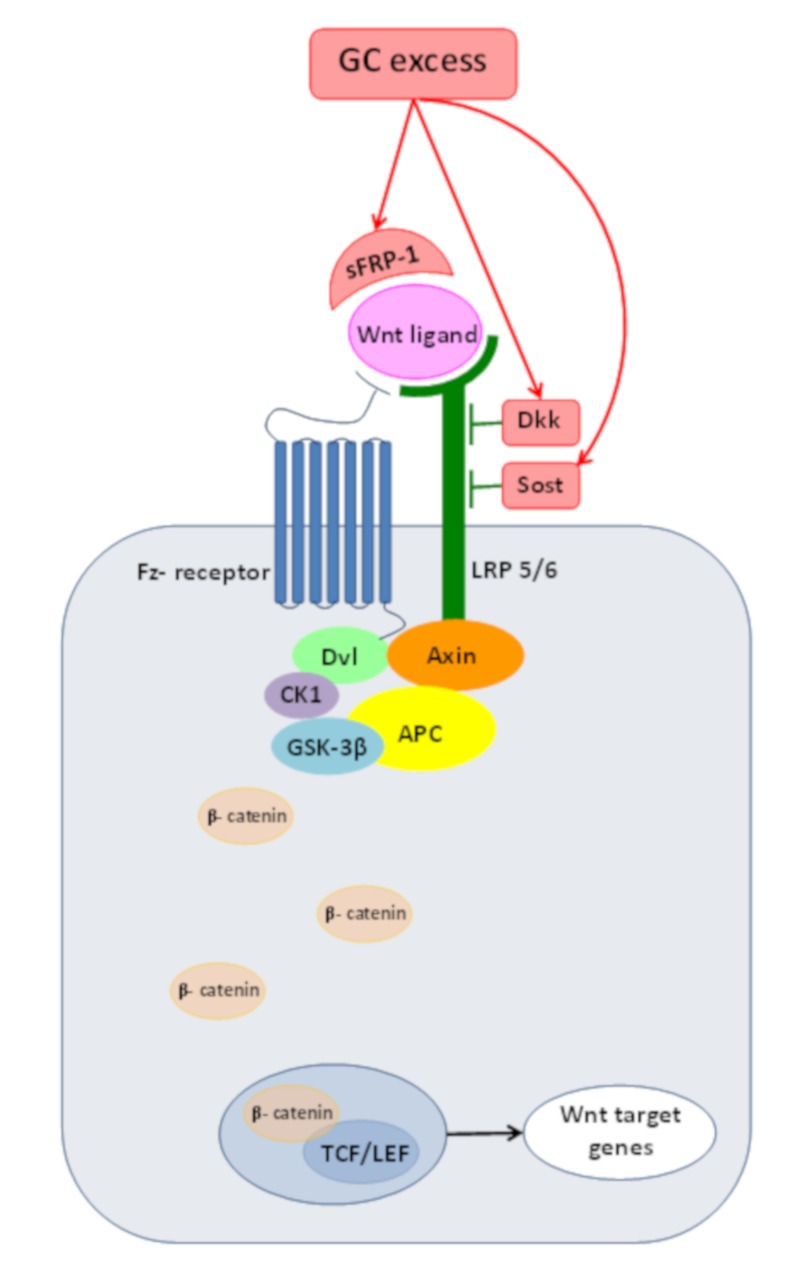
Schematic representation of the Wnt/β-catenin pathway in the presence of a Wnt-ligand. Glucocorticoid (GC) excess leads to the accumulation of Wnt-inhibitors: Dkk-1, sclerostin (Sost) and secreted Frizzled-related protein-1 (s-FRP-1). Dkk-1 inhibits Wnt-signaling through LRP 6 [11], sclerostin exerts its inhibitory role by binding to the extracellular domain of LRP 5 [12] and sFRP-1 modulate Wnt-signaling by binding to the Wnt-ligand and preventing Wnt-receptor activation [13].

**Figure 2 molecules-25-01489-f002:**
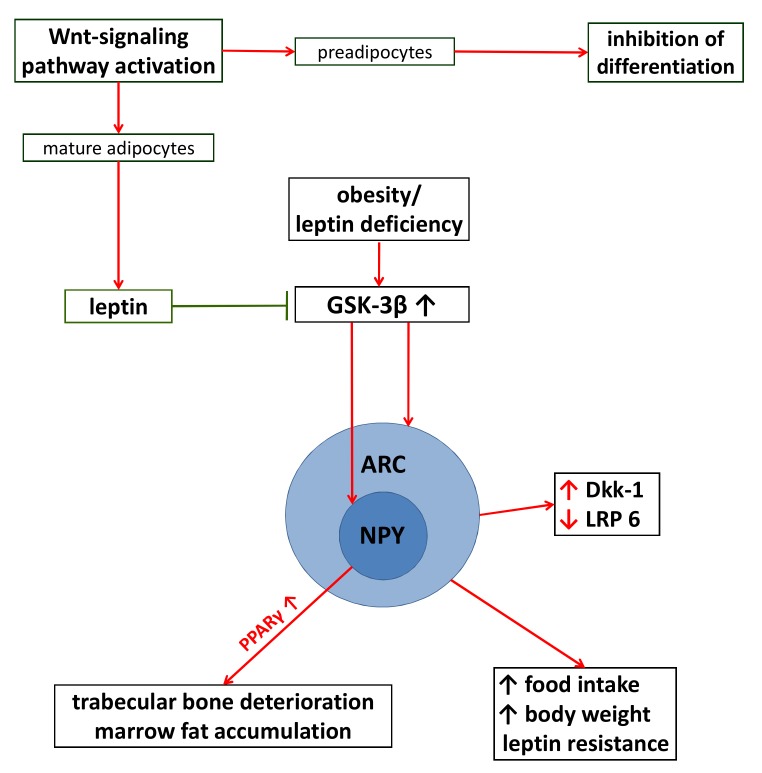
The interaction between adipose tissue and the Wnt/β-catenin pathway. Wnt-signaling activation represses preadipocytes’ differentiation, but stimulates leptin production in mature adipocytes [61]. Genes involved in Wnt-pathway are mostly expressed in mediobasal hypothalamic cells with particular prominence in the arcuate nucleus (ARC) [62]. Obesity and leptin deficiency in mice leads to increased GSK-3β activity in ARC, accompanied by increased Dkk-1 and decreased LRP 6 levels; this status leads to hyperphagia and a consequential aggravation of glucose intolerance, obesity and leptin resistance [62]. Increased GSK-3β activity in neuropeptide Y (NPY) cells leads to bone deteriorations and marrow fat accumulation in mice; NPY deletion is protective against glucocorticoid-induced bone mass loss and fatty marrow development (through PPARγ modification) [64]. Leptin treatment fully restores decreased LRP 6 activation, enhances GSK-3β inactivation in both ARC and NPY cells [62,63].

**Figure 3 molecules-25-01489-f003:**
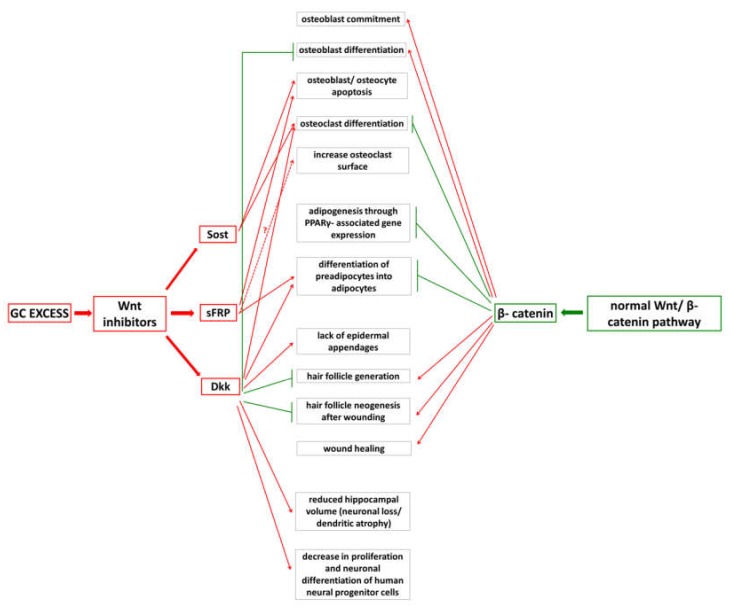
Schematic representation of the interplay between GC-excess and Wnt/ β-catenin pathway.

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
