# Peer review of "Glucocorticoids Influencing Wnt/β-Catenin Pathway; Multiple Sites, Heterogeneous Effects"

_molecules, 2020, doi:10.3390/molecules25071489_

Round 1

Reviewer 1 Report

 This paper is well written and designed, directed to the effect of glucocorticoids excess on the inhibition of Wnt-pathway in adipous tissue, brain, skin. Stimulating effect of exercise and hypocaloric diet on creating sufficient hormonal environment were ostoblasts restored their function is described.This paper have  theoretical and practical value. 

Author Response

We are gratefull for the Reviewer for his/her positive comments.

Reviewer 2 Report

There are many places in the text where editing of the English is required, the abstract in particular. As an example, this reviewer examined the first page of the manuscript and have found that the lines listed below need to be addressed, and acknowledges that there may be more.

Line 10- reword ("tight" is incorrect)

Line 12- reword ("comprehensive" is incorrect)

Line 13- reword, meaning is not quite correct (and "hole" should be "whole")

Line 14- ..."a" relatively rare condition...

Line 15- reword, syntax not correct

Line 17- while "with special regard" is correct here, this is an unusual usage of this phrase

Line 24- "its" not "it's"

Line 28- "pathways" not "pathway"

Line 34- "Its" not "It's"

Line 41- ...to "the" Fz receptor...

Line 43- "being binded" should read "being bound"

Author Response

We thank the Reviewer for his/her comments.

Extensive language editing and corrections have been made through the manuscript. All changes are highlighted in yellow in our revised manuscript.

Reviewer 3 Report

The Review article by Meszaros et al., makes a compelling discussion on the role of GC influencing bone, adipose tissue , brain and the skin with emphasis on the Wnt/ß catenin signaling. Notable emphasis has been placed on cushing's syndrome and its associated affected organs. More figures and tables that very well align with the review title are needed, fig 1 is a good example, but table 1 and 2 do not provided much information. While the authors very well describe the GC excess condition and the homeostatic role of Wnt signaling in this process, they do require a better flow for easy reading, which has been difficult in certain sections as highlighted below.  

1.Abstract: Paragraph 1 and 2 seem redundant and can be made concise and crisp to narrate what the review holds.

Line 13: edit hole

Repeat sentences such as stated below should be avoided within such a small section

Line 11 and 19:'GC regulate many physiological process' 

Line 17 and 29: repeat sentences on what the review will focus on : bone, adipose, bain and skin

2. Line 43: please edit 'being binded' to bound

3. Section 2: restructuring introduction to first introducing Cushing's disease from line 66 and then bringing in Wnt's role in this disease  line 52 would make this an easy read. 

Table 1 : This table could be made more informative and relevant to the title.

Please add in reference number against each row, so the audience can have a straight forward access to papers of interest. This is also applicable since the % reported is based on total subjects included in individual studies and not overall general % of occurrence as a cumulative finding from various studies for the disease. Please confirm in the text as this could be misleading.

Data on males and females in this study would be informative

Information on what is known about Wnt signaling (signaling components) with references, in each of these conditions will be useful.

4. Section 3 and 4 can be easily merged, as they highlight GC excess, Wnt signaling and the bone

The paragraphs following up in these 2 sections do not flow easily and should be interconnected, have had to go back and forth within this section as the paragraphs seem disparate

4.1 and 4.2 could be merged as mechanisms of GC effect on the bone

Section 4.4 could followup post Table 2, to maintain easy reading and role of Wnt on bone cytokines

Table 2. The legend and the table seem redundant and the text seems more explanatory. Keeping either/or is suggested.

Title for 4 could be more explanatory since it also included Wnt signaling information and not just GC excess on the bone.

5. Sections 5, 6, 7 could be broken into subsections like the previous 2 sections have been to maintain consistency. Some figure or table describing these sections would make the review easy.

6. Line 467: edit occurs

7. Line 190: edit develops

Author Response

We are grateful for this Reviewer for his/her positive comments and suggestions. All changes are highlighted in yellow in our revised manuscript. Below, please find our point-to point answers.

The Review article by Meszaros et al., makes a compelling discussion on the role of GC influencing bone, adipose tissue , brain and the skin with emphasis on the Wnt/ß catenin signaling. Notable emphasis has been placed on cushing's syndrome and its associated affected organs. More figures and tables that very well align with the review title are needed, fig 1 is a good example, but table 1 and 2 do not provided much information. While the authors very well describe the GC excess condition and the homeostatic role of Wnt signaling in this process, they do require a better flow for easy reading, which has been difficult in certain sections as highlighted below.  

Answer: According to the Reviewer's comments we restructured our manuscript .

1.Abstract: Paragraph 1 and 2 seem redundant and can be made concise and crisp to narrate what the review holds.

Answer: it was corrected.

Line 13: edit hole, it was corrected.

Repeat sentences such as stated below should be avoided within such a small section.

Answer: redundant sentences have been deleted from the revised manuscript.

Line 11 and 19:'GC regulate many physiological process' : It was corrected.

Line 17 and 29: repeat sentences on what the review will focus on : bone, adipose, bain and skin.

It was corrected

2. Line 43: please edit 'being binded' to bound, It was corrected

3. Section 2: restructuring introduction to first introducing Cushing's disease from line 66 and then bringing in Wnt's role in this disease  line 52 would make this an easy read. 

Table 1 : This table could be made more informative and relevant to the title.

Please add in reference number against each row, so the audience can have a straight forward access to papers of interest. This is also applicable since the % reported is based on total subjects included in individual studies and not overall general % of occurrence as a cumulative finding from various studies for the disease. Please confirm in the text as this could be misleading.

Data on males and females in this study would be informative

Information on what is known about Wnt signaling (signaling components) with references, in each of these conditions will be useful.

Answer: the Table was deleted from the revised manuscript. All data about the prevalence of signs and symptoms have been incorporated into the relevant sections of the manuscript.

4. Section 3 and 4 can be easily merged, as they highlight GC excess, Wnt signaling and the bone

The paragraphs following up in these 2 sections do not flow easily and should be interconnected, have had to go back and forth within this section as the paragraphs seem disparate

4.1 and 4.2 could be merged as mechanisms of GC effect on the bone

Section 4.4 could followup post Table 2, to maintain easy reading and role of Wnt on bone cytokines

Table 2. The legend and the table seem redundant and the text seems more explanatory. Keeping either/or is suggested.

Title for 4 could be more explanatory since it also included Wnt signaling information and not just GC excess on the bone.

All these sub-chapters have been restructured.

5. Sections 5, 6, 7 could be broken into subsections like the previous 2 sections have been to maintain consistency. Some figure or table describing these sections would make the review easy.

Based on this suggestion subsections were made.

6. Line 467: edit occurs

It was edited.

7. Line 190: edit develops

Round 2

Reviewer 2 Report

This manuscript is significantly improved.